# Matricellular Proteins in the Homeostasis, Regeneration, and Aging of Skin

**DOI:** 10.3390/ijms241814274

**Published:** 2023-09-19

**Authors:** Erna Raja, Maria Thea Rane Dela Cruz Clarin, Hiromi Yanagisawa

**Affiliations:** 1Life Science Center for Survival Dynamics, Tsukuba Advanced Research Alliance (TARA), University of Tsukuba, Tsukuba 305-8577, Japan; rajaerna@tara.tsukuba.ac.jp (E.R.); s2130537@u.tsukuba.ac.jp (M.T.R.D.C.C.); 2Ph.D. Program in Humanics, School of Integrative and Global Majors (SIGMA), University of Tsukuba, Tsukuba 305-8577, Japan

**Keywords:** matricellular proteins, basement membrane, extracellular matrix, epidermal stem cells, fibroblasts, skin aging, wound healing, skin regeneration

## Abstract

Matricellular proteins are secreted extracellular proteins that bear no primary structural functions but play crucial roles in tissue remodeling during development, homeostasis, and aging. Despite their low expression after birth, matricellular proteins within skin compartments support the structural function of many extracellular matrix proteins, such as collagens. In this review, we summarize the function of matricellular proteins in skin stem cell niches that influence stem cells’ fate and self-renewal ability. In the epidermal stem cell niche, fibulin 7 promotes epidermal stem cells’ heterogeneity and fitness into old age, and the transforming growth factor-β—induced protein ig-h3 (TGFBI)—enhances epidermal stem cell growth and wound healing. In the hair follicle stem cell niche, matricellular proteins such as periostin, tenascin C, SPARC, fibulin 1, CCN2, and R-Spondin 2 and 3 modulate stem cell activity during the hair cycle and may stabilize arrector pili muscle attachment to the hair follicle during piloerections (goosebumps). In skin wound healing, matricellular proteins are upregulated, and their functions have been examined in various gain-and-loss-of-function studies. However, much remains unknown concerning whether these proteins modulate skin stem cell behavior, plasticity, or cell–cell communications during wound healing and aging, leaving a new avenue for future studies.

## 1. Introduction

Matricellular proteins are nonstructural, modular, extracellular proteins that exert their effects by binding to cell surface receptors, extracellular matrix (ECM) proteins, soluble signaling molecules, and proteases, thereby modulating cellular responses to changes in their microenvironment, particularly during tissue remodeling [1,2]. The term matricellular protein was first introduced by Paul Bornstein in the 1990s to describe thrombospondin 1 (TSP1/THBS1), secreted protein acidic rich in cysteine (SPARC), and tenascin C [1]. Since then, the list has substantially expanded and includes protein families of the cellular communication network (CCN) factors, the fasciclin family, tenascins, galectins, R-Spondins (RSPOs), fibulins, ecto-nucleotide pyrophosphatase/phosphodiesterases, small integrin-binding ligand N-linked glycoproteins, and olfactomedins [3]. Although matricellular proteins have no direct structural functions, they can bind to structural ECM proteins such as collagens [4,5]. However, some structural ECM proteins have nonstructural functions, such as fibrillin-1 [6] and fibronectin [7], which bind and regulate the bioavailability of growth factors in the tissue microenvironment. Generally, matricellular proteins are highly expressed during development, which decreases to low levels during homeostasis but is upregulated upon tissue injury and disease [4]. While some knockout animal models of matricellular proteins with developmental roles have exhibited embryonic or neonatal lethality [8,9,10,11], others have resulted in no apparent post-natal phenotypic changes, and their biological functions were later discovered during tissue damage or in other pathological contexts [12,13,14,15]. In this review, we summarize reports on matricellular proteins in adult skin homeostasis, regeneration, and aging. We focus on the interaction between matricellular proteins and tissue stem cells and how they modify the tissue stem cell environment.

## 2. Skin Architecture, ECM, and Cellular Components

The skin is the largest organ of the body and protects us against environmental insults. It shields the body from mechanical abrasion, pathological infections, dehydration, and fluctuations in body temperature, while the nerves in the skin also provide us with sensations of touch [16]. The skin needs to act as a resilient mechanical barrier, yet provide structural flexibility. The functional unit of skin consists of the stratified epidermis and dermis (including dermal adipose and skin appendages such as hair follicles, sweat, and sebaceous glands) as well as the panniculus carnosus (PC) muscle and the subcutaneous fascia (Figure 1). Notably, the human skin has a thicker epidermis and dermis compared with mouse skin, and the epidermis exhibits undulations forming the rete ridge and inter-ridge (also known as dermal papillae) structures that are absent in mouse skin (Figure 1).

### 2.1. Epidermis

The construction of the epidermis begins with the basal epidermal stem cells that undergo a continuous and balanced process of symmetric self-renewal and asymmetric division to produce progenitors and differentiated cells. These provide a constant supply of keratinocytes committed to terminal differentiation which form the suprabasal layers and the cornified skin barrier [61]. Through hemidesmosomes and integrin receptors, the basal epidermal stem cells are anchored to the basement membrane (BM), which connects the epidermis to the underlying dermis. The BM’s primary constituents are structural scaffolding matrix proteins such as collagens IV, VII, XVII, laminin 332, and laminin 511 [62]. Fibronectin is also present in the lamina lucida of the BM (the laminin-rich area) [63]. Matricellular proteins, such as fibulin 2 [17,64], fibulin 7 [18], SPARC [19,20], hemicentin 1 (fibulin 6) [22,23], THBS1 [24], and thrombospondin 5 (THBS5 or COMP) [65], are contained in the epidermal BM via interactions with structural proteins or the integrin receptors.

Functionally, fibulin 2 binds to laminin 332 to stabilize the epidermal BM in neonatal skin. The absence of fibulin 2 results in separation of the dermal–epidermal junction (DEJ) and skin blisters [17]. Fibulin 7 is also expressed in the epidermal BM and has been shown to bind to collagen IV in vitro [18], although its role in the BM arrangement remains unknown. SPARC was similarly shown to bind to collagen IV in vitro and induce collagen IV and VII expression in the epidermal BM in reconstructed human three-dimensional skin culture models [19,20,21]. Hemicentin 1 co-localizes with laminin α2 at the epidermal BM [22]. *Hmcn1* null mice exhibit unevenly widened lamina lucida and lamina densa and compromised hemidesmosomes [22], which suggests its function in BM organization. Hemicentin 1 was further shown to compete with laminins for its binding site to the nidogen 2 proteoglycan during BM formation and maintenance [23]. In contrast to other matricellular proteins in the BM, THBS1 is believed to act as an endogenous angiogenesis inhibitor, forming a barrier between the nonvascular epidermis and the vascularized dermis [25,26]. Matricellular proteins at the BM/DEJ, such as fibulin 2 and SPARC, are produced by both keratinocytes and fibroblasts [17,66,67,68,69], with hemicentin 1 and COMP mainly being secreted by fibroblasts [22,70]. In contrast, a subset of basal keratinocytes expresses *Thbs1*, albeit at low abundance during homeostasis [71]

In addition to keratinocytes, the epidermis is home to resident immune cells and melanocytes, which provide pigmentation to the skin and hair [16]. Melanocytes are regenerated by melanocyte stem cells, which are neural-crest-derived cells. Melanocytes produce melanin and have long dendrites that can make physical connections with up to 40 keratinocytes. Melanin is transferred to keratinocytes via caveolae-dependent internalization (lipid-raft-mediated endocytosis) to protect their nuclei from ultraviolet (UV) radiation [72]. Melanocyte survival also depends on its attachment to the epidermal BM via binding between its discoidin domain receptor 1 (DDR1) and collagen IV [27]. During stress or UV radiation, keratinocytes secrete paracrine factors such as IL-1β that, in turn, induce melanocytes to secrete the matricellular protein CCN3. CCN3 promotes DDR1 binding to collagen IV and inhibits melanocyte proliferation [27,28].

### 2.2. Dermis

Underneath the epidermal BM are the papillary and reticular dermal compartments, which are distinct in their cellularization and ECM components (Figure 1) [73]. The upper dermis/papillary dermis is densely populated with papillary fibroblasts that reside within thin and loose networks of fibrillar collagens (I and III) and elastic fibers. They produce ECMs such as collagen VI [74], fibronectin, and the matricellular proteins tenascin C and fibulin 2 [66]. These ECM proteins interact with each other to maintain structural integrity and provide biological cues for the surrounding cells. Fibronectin is a glycoprotein required for the formation of microfibrils. It acts as a scaffold for the elastic fibers, modulates the balance of skin rigidity and elasticity, and supports cell attachment and migration along the ECM matrix, such as collagen fibers [75,76]. Fibronectin also possesses growth-factor-binding ability [35,36]. Collagen VI fibrils interact with both fibronectin and collagen I [77] and may promote the tensile strength of the skin [78]. While fibulin 2 binds to fibrillin 1 and regulates the homeostasis of elastic fibers [37,38], further maintenance of the dermal ECM may come from the binding of tenascin C to fibronectin, collagen, and periostin [29]. Tenascin C has been reported to control cell proliferation by acting as a constitutive ligand to activate the cell surface epidermal growth factor receptor [30,31] and by sequestration of soluble growth factors such as Wnt3a, TGFβ, and VEGF [29,32,33,34]. Similarly, periostin is expressed in the papillary dermis, interacts with fibronectin, collagen I, and tenascin C [79] and is reported to modulate collagen structure and stability [43]. Periostin also impacts keratinocyte proliferation via fibroblast paracrine secretion of IL-6 [44].

The softer papillary dermis serves as a cushion connecting the stiffer epidermal BM to the lower reticular dermis, a major part of the dermal compartment. Reticular dermis contains more sparsely distributed reticular fibroblasts, thick and highly organized collagen bundles that contribute to skin tensile strength [62,73]. Reticular fibroblasts produce ECMs such as collagen I, fibrillin 1, fibronectin, emilin 1, and the matricellular proteins THBS1, tenascin C, and fibulin 2 [66]. Fibrillin 1, elastin, emilin 1, fibronectin, and matricellular proteins such as fibulin 2, 4, and 5 are expressed in the dermis and crucial for the formation of elastic fibers, which provide the skin with elastic properties [39,40,41,42,80]. Aside from regulating the balance of dermal vascularization, THBS1 is physically associated with the collagen I KGHR motif, which is important for cross-linking. The loss of this interaction results in abnormal collagen I and human dermal fibroblast differentiation into myofibroblasts [45]. Similarly, *Thbs2* knockout mice have disarranged collagen fibril sizes and patterns in the dermis, which correlate with reduced skin tensile strength [46]. Abnormal collagen fibrils have also been observed in SPARC null mice, which exhibit decreases in the amount of dermal collagen I, the fibril diameter, and the tensile strength [47].

Fibroblasts in the embryonic/neonatal mouse skin are proliferative. However, in the mature skin, fibroblasts cease to divide and produce more ECM in the dermis. This ECM (collagen) enrichment in the adult skin acts as negative feedback to further inhibit fibroblast proliferation, although this quiescent state is reversible and fibroblasts are activated again upon skin injury [81]. Resident dermal immune cells are also activated to produce various cytokines that modulate fibroblast differentiation and proliferation [81,82]. Dermal fibroblasts originate from a common multipotent mesenchymal cell progenitor (expressing Pdgfrα, Dlk1, and Lrig1) that give rise to papillary and reticular fibroblast progenitors [82].

### 2.3. Dermal Adipose Tissue

Reticular fibroblast progenitors can give rise to adipocyte precursors and mature, lipid-filled adipocytes [82,83], which constitute the dermal fat layer. Dermal adipose functions as energy storage, thermal insulation, and mechanical support. Furthermore, it harbors innate immune antimicrobial functions [84,85]. Mature adipocytes express the collagen IV BM protein, which is deposited pericellularly and is believed to act as a strong scaffold to protect cells from mechanical stress in this loose connective tissue [86,87]. Collagen IV expression around these adipocytes is increased in obese compared with lean human subcutaneous adipose tissue [88], possibly to counterbalance the physical changes associated with an increase in adipocyte cell size. Other ECMs associated with skin adipose are collagen I, III, V [89], and VI [90]. Proteoglycans such as versican, biglycan, and decorin are also expressed in adipose tissue in order to bind to collagens, support their scaffolding function, and counteract compressive forces. However, in cases of obesity, these proteoglycans are present in excessive amounts, resulting in ECM defects and the promotion of tissue inflammation [91,92,93]. Adipogenesis is influenced by the matricellular protein SPARC, as its loss of expression leads to increased dermal adipose tissue [48] and accelerated wound healing [49]. Similarly, the secreted protein coiled-coil domain containing 80 (CCDC80/URB/DRO1) is highly expressed by adipocytes to modulate adipogenesis, and its genetic deletion in mice promotes body fat deposition, including subcutaneous fat [50,51,52].

### 2.4. The Hair Follicle

The hair follicle structure spans from the hypodermis (dermal adipose) up to the epidermis. Its size changes according to the hair cycle, which consists of the telogen (resting), anagen (growing), and catagen (regression) phases [94]. Hair follicles are connected to the sebaceous glands, arrector pili muscles, and dermal papilla at the base of the follicle (Figure 1 and Figure 2). During homeostasis, the hair follicle is mainly regenerated from hair follicle stem cells in the bulge region, with coordination from signals originating from mesenchymal cells in the dermal papilla, which control the exit from telogen and the duration of anagen [95,96,97,98,99,100] (Figure 2). The dermal papilla cells are, in turn, replenished by a subset of dermal sheath cells, which have self-renewal ability, also referred to as hair follicle dermal stem cells [96]. Hair follicle BM proteins are expressed by both epithelial cells and fibroblasts, with distinct contributions [101]. They consist of core structural BM proteins such as laminins, collagens IV, VI, VII, XVII, and XVIII, netrins 1 and 4, fibronectin, nephronectin, and matricellular proteins such as periostin, tenascin C, fibulin 1, SPARC, SMOC1, spondin-1, and R-spondin 3 [75,101,102]. The function of fibulin 1 is likely related to the regulation of BM homeostasis through its binding to laminin and collagen IV [53,103]. Nephronectin is produced by hair follicle stem cells to form an adhesion point for the arrector pili muscle cells expressing α8β1 integrins. Arrector pili muscles and the sympathetic nervous system cooperate with hair follicles to achieve piloerections to keep warm air closer to the skin in response to cold temperatures or emotions [54]. It has been proposed that hair follicle stem cells expressing periostin, tenascin C, and SPARC resemble tendon-related gene functions, i.e., they are expressed in the tendonous part of muscle tissue, which strengthens the bone–muscle connection and allows for skeletal movement [55,56,57]. Similarly, their deposition in the hair bulge matrix may stabilize the connection between arrector pili muscles and hair follicles during piloerections [54].

**Figure 2 ijms-24-14274-f002:**
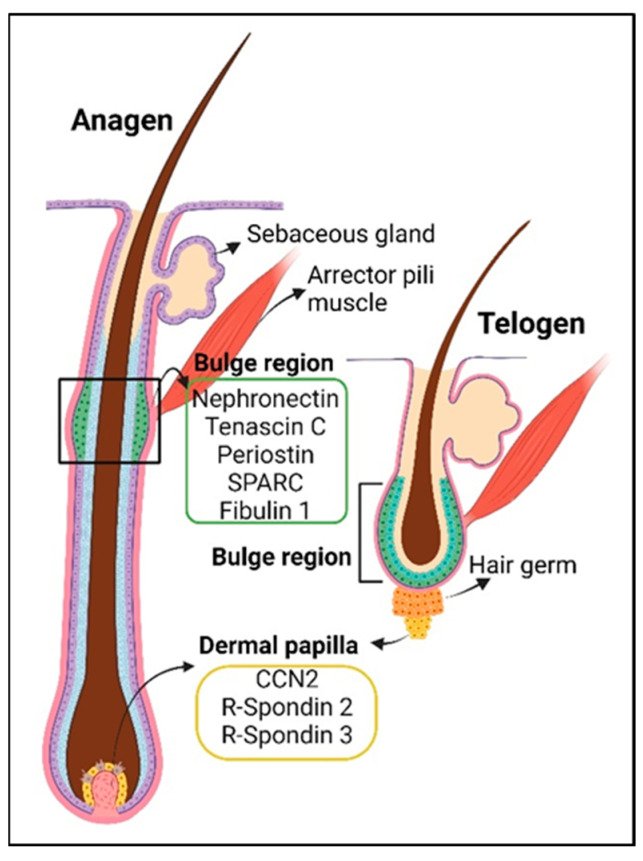
The hair follicle in anagen (growth phase, left) and telogen (resting phase, right) and the associated matricellular proteins in the hair follicle stem cell niche (bulge region, green box) and dermal papilla niche (yellow box). The functions of these matricellular proteins are summarized in Table 1 and Table 2. Adapted from BioRender.com (accessed on 2 August 2023).

### 2.5. The Sebaceous Gland

Sebaceous glands are located in the upper and permanent area of the hair follicle (junctional zone) in hair-associated skin. However, they may also be present in the skin independently of hair follicles such as in the meibomian glands of the eyelid [104,105]. The peripheral (basal) epidermal cells of these glands are proliferative and contain sebaceous gland stem cells. The differentiated sebocytes reside in the inner layer of the gland. Through cell death and lysis, lipids/sebum are released onto the skin surface to promote barrier functions, water repulsion, and antimicrobial- and antioxidant activities (vitamin E) [105,106,107]. To our knowledge, documentation regarding the ECM of the sebaceous gland is limited. A recent report indicated expression and interaction of fibronectin with the sebaceous gland basal cells to regulate its differentiation [58]. Another report discussed the ECM in the meibomian glands such as collagen IV, laminin α2, and β1, which may serve as BM scaffolding proteins, and the matricellular protein tenascin C, which may modulate growth factor bioavailability around sebaceous gland stem cells [59].

**Table 2 ijms-24-14274-t002:** Matricellular proteins for maintenance of hair follicles and sebaceous gland stem cells.

Process	Matricellular Protein	Function	References & Study Model
Hair follicle stem cells homeostasis	Tenascin C	promotes WNT signaling and maintains hair follicle stem cells	[32] (mouse)
	Periostin	pro-cell proliferation in the hair follicles post-wounding	[108] (mouse)
	CCN2	maintenance of hair follicle stem cell quiescence	[109] (mouse)
	R-spondin 2	activates hair follicle stem cells, cell proliferation of dermal papilla and dermal sheath	[110] (human hair follicles & mouse), [111] (human scalp & mouse skin, [96] (mouse)
	R-spondin 3	cell proliferation of dermal papilla and dermal sheath	[96] (mouse), [111] (human scalp & mouse skin)
	Mindin (Spon2)	gene expression is decreased in aging hair follicle stem cells, role unknown in the hair follicle	[112] (mouse)
	Thrombospondin 1	gene expression is increased in aging hair follicle stem cells, hair follicle and vasculature regression in catagen	[112] (mouse skin), [113] (mouse)
Sebaceous gland stem cells	Fibronectin *	maintenance of basal sebocytes	[58] (mouse)

* Fibronectin is not classified as matricellular proteins but they bear non-structural functions too via binding to matricellular proteins or regulating growth factors’ bio-availability.

### 2.6. The Panniculus Carnosus Muscle

The PC muscle is a layer of skeletal muscle located directly under the dermal adipose tissue in some mammals to facilitate skin movement independently of deeper muscle mass. The PC muscle has been suggested to mediate skin twitching during irritation, facial skin movement for social expressions, shivering thermogenesis, and skin contraction to promote wound closure [114]. In humans, PC muscle remnants exist only in certain anatomical regions, such as the craniofacial muscle, the platysma muscle (ventral region of the neck), and the palmaris brevis in the hand [114]. The PC muscle is reported to be highly regenerative [115] and vascularized [116]. The fibulin 4 matricellular protein may regulate PC muscle homeostasis, as the *Fbln4* E57K homozygous mutation (the equivalent of the human mutation causing cutis laxa) in mice exhibits a thinner dermis and PC muscle layer, possibly due to elastic fiber defects and abnormal collagen fibrils [60]. However, it is unknown whether this mutation compromises PC muscle function.

### 2.7. Subcutaneous Fascia

The subcutaneous fascia ECM is enriched in an aqueous matrix containing glycosaminoglycans such as hyaluronan to facilitate smooth gliding between the skin and muscle. Fibroblasts and fasciacytes contribute to ECM production in the fascia [117]. Fibroblasts secrete fibrous matrix components such as fibrillin, elastin, fibronectin, and collagens I and III, whereas hyaluronan is produced by fasciacytes [118]. Fasciacytes are round, fibroblast-like cells expressing vimentin [119]. Little is known about the role of matricellular proteins in subcutaneous fascia maintenance, although the loss of fibulin 3 compromises elastic fibers in the visceral fascia [120,121]. Interestingly, lineage tracing experiments have demonstrated that, during wound healing, fascia fibroblasts are mainly responsible for generating scar tissue and not dermal fibroblasts [122,123].

## 3. Matricellular Proteins in Stem Cell Regulation during Homeostasis, Injury Repair, and Chronological Skin Aging

During physiological wear and tear or tissue injury, adult skin stem cells maintain tissue functions by replenishing damaged cells. Activated stem cells proliferate to self-renew and generate committed progenitor cells, forming terminally differentiated cells with specified functions. Stem cell fate regulation is influenced by changes in its microenvironment [112,124,125]. Through mechanotransduction pathways involving crucial players such as integrin receptors binding to the ECM, Piezo1 calcium channel, and YAP/TAZ signaling, both stem cells and stromal cells sense changes in the biochemical and mechanical properties of their microenvironment and respond accordingly [75,124,126,127,128]. The ECM/matricellular proteins are components of the microenvironment and changes in their quality and quantity with aging compromise skin functions.

In aged skin, the epidermis is thinner and less resistant to shear stress, and the rete ridge structures are compromised [129]. Moreover, hair follicles are miniaturized or depleted [130], while dermal fibroblasts lose their ability to produce collagen, exhibit altered identity [131], and become more adipogenic [132]. Immunopathological changes also occur in the dermis, and overall wound healing ability is impaired [129]. These declining skin features are partly attributed to impaired stem cell functions due to stem cell depletion or changes in their fate/behaviors. Furthermore, as skin ages, proliferating stem cells are prone to accumulate DNA damage due to intrinsic replication stress and exposure to UV irradiation over time. ECM stiffness may determine cells’ response of repairing double-strand DNA breaks, which can lead to genetic instability. Low ECM stiffness weakens the double-strand DNA break repair mechanism and renders cells more sensitive to genotoxic stress [133]. In this section, we will discuss the reported roles of matricellular proteins in maintaining stem cell homeostasis, changes during injury-induced regeneration, and aged skin.

### 3.1. Epidermal Stem Cells

Epidermal stem cells (more specifically interfollicular epidermal stem cells) that form stratified skin epithelium comprise heterogeneous populations in mouse [134] and human skin [135,136] (Figure 3). They differ in their proliferative capacity (hence the names: slow- and fast-cycling stem cell populations) and in their molecular characteristics [18,134,136,137]. In the mouse tail skin model, these two populations differentiate to form distinct territories, with fast-cycling stem cells replenishing the scale region, whereas the slow-cycling stem cells maintain the interscale region [134]. Interestingly, the gene expressions of the slow- and fast-cycling stem cells in the mouse tail skin also show territorial expression patterns that reflect the two basal epidermal stem cell regions of human skin, namely the inter-ridge (slow-cycling region) and rete ridge (fast-cycling region) [135]. An early study using monkey palm skin demonstrated that tritiated thymidine-labelled cells are located at the tips of the rete ridges [136], where basal cells express fast-cycling stem cell marker genes [135]. This suggests that the epidermal stem cell heterogeneity model is conserved [138,139].

Several signaling pathways organize the territorial segregation of the slow- and fast-cycling stem cells (in the interscale and the scale of tail skin), including Wnt, Lrig1, and Edaradd [140,141]. ECM, such as collagen XVII, a hemidesmosome component expressed by epidermal stem cells, is required for proper scale patterning [142], possibly through positive regulation of Wnt signaling [143,144]. The distinctive characteristics between slow- and fast-cycling stem cells raise intriguing questions about their biological relevance. It has been suggested that such heterogeneity confers robustness to skin homeostasis during environmental challenges such as UV exposure [135]. Interestingly, these two stem cell populations become interchangeable during wound repair, although they return to their initial territories once the wound is healed [134]. It is unknown whether the induction of matricellular protein expression during injury repair modulates the behavior of the two stem cell types.

#### 3.1.1. Keratinocyte Regulation during Injury-Induced Skin Regeneration

Skin wound healing is an intricate process involving the interactions of many ECMs, growth factors, and various cell types [3,145,146]. It begins with the injury site filling with blood clotting factors and chemokines to attract leukocytes, including neutrophils, to clean the wounded area. Macrophages then enter the area to phagocytize neutrophils and secrete growth factors to stimulate the activation of keratinocytes and fibroblasts, resulting in ECM and BM production, re-epithelialization, and granulation tissue formation accompanied by neo-vascularization and tissue remodeling [146]. Here, we focus on the effect of matricellular proteins during the re-epithelialization process (Table 3), where keratinocytes and multiple skin stem cells proliferate and migrate to close the wounded area.

**Table 3 ijms-24-14274-t003:** Matricellular proteins regulating wound healing (re-epithelialization), epidermal stem cells, and aging skin.

Process	Matricellular Protein	Function	References & Study Model
Wound healing (Re-epithelialization)	CCN1	upregulated during skin repair, induces myofibroblast senescence to minimize fibrosis, enhances neutrophil efferocytosis by macrophages for resolution of inflammation	[147,148] (summary), [149,150] (mouse)
		promotes keratinocytes proliferation and induces EMT during skin expansion	[151] (human primary keratinocytes & in vivo mouse), [152] (human & rat skin)
	Thrombospondin 1	spatiotemporal expression regulates wound healing processes	[153,154] (mouse)
		promotes keratinocyte migration and proliferation	[71] (human keratinocytes culture & in vivo mouse), [155] (mouse)
	Thrombospondin 2	influences vascularization but not re-epithelialization	[153,156] (mouse)
	Thrombospondin 4	accelerates wound healing through increased keratinocytes proliferation and fibroblasts migration	[157] (human & mouse skin)
	Thrombospondin 5	suppresses keratinocyte activation	[65] (human skin)
	Periostin	regulates re-epithelialization and wound contraction, downregulated in aging skin	[108,158,159] (mouse), [43] (human skin)
	R-spondin 1	regulates skin differentiation and malignancy, promotes wound healing and re-epithelialization	[160] (human & mouse), [161] (rat)
	TGFBI	BM component, promotes epidermal stem cell proliferation and re-epithelialization	[162] (human & mouse)
Aging	Fibulin 7	decreased in aged BM, maintains epidermal stem cell heterogeneity, binds to periostin and tenascin C and improves re-epithelialization in aging skin wound healing	[18] (mouse)
	Thrombospondin 1	upregulated in epidermal stem cells and dermal fibroblasts during aging and may contribute to aging-associated vasculature changes	[124,163] (mouse), [164] (human dermal fibroblasts)
	Mindin (Spon2)	Mindin promotes epidermal stem/progenitor cell expansion downstream of Snail, inhibition of Mindin and fibulin 5 rescues Snail-induced skin fibrosis	[165,166,167] (mouse)
	Fibulin 5		[168] (mouse)
	CCN1	promotes fibroblast senescence in skin dermis	[149,169] (mouse)
	CCN2	decreased in aging dermis is associated with collagen loss	[170] (human skin & dermal fibroblasts), [171] (human dermal fibroblasts)

The CCN1 matricellular protein is upregulated during skin repair [147] and was initially found to promote wound healing by inducing myofibroblast senescence to minimize fibrosis [149] and by enhancing neutrophil efferocytosis by macrophages to stimulate inflammatory resolution [150]. Later, CCN1 was found to support re-epithelialization by promoting keratinocyte proliferation and migration [151]. Intriguingly, CCN1 has also been shown to increase in the epidermis during skin expansion in humans and rats, inducing proliferation and the epithelial-to-mesenchymal transition (EMT) [152,172], a reversible and temporary process whereby epithelial cells gain mesenchymal properties that contribute to increasing cell migration and re-epithelialization.

*Thbs1* knockout mice exhibit delayed wound healing characterized by persistent inflammation, and reduced macrophage infiltration and TGFβ1 expression [153]. However, constitutive overexpression of *Thbs1* under the keratin 14 promoter has also resulted in delayed re-epithelialization and overall wound healing due to defective granulation tissue and the inhibition of wound angiogenesis [26,154]. These observations underscore the importance of spatiotemporal regulation of *Thbs1* expression in wound healing. Moreover, a recent report using single-cell RNA sequencing indicated that basal keratinocytes at the migrating front highly express *Thbs1* [71,155]. These findings were supported by immunostainings and in vitro studies describing THBS1′s role in promoting keratinocyte migration and proliferation [71]. In contrast, *Thbs2* deletion results in accelerated wound healing in mice without changes in the re-epithelialization rate due to higher vascularization, although the newly formed epidermis in *Thbs2* null mice is thicker and forms an unusual rete ridge structure similar to the human skin architecture [153,156]. THBS4 is also upregulated in inflamed human psoriatic skin, post-burn injuries, and mouse dorsal skin wound assays. The administration of recombinant THBS4 to wounded skin in mice has led to faster wound healing, possibly due to increased keratinocyte proliferation and fibroblast migration [157]. Furthermore, COMP/THBS5 co-localizes with laminin α1 and integrin β1 at the DEJ, where it was found to suppress keratinocyte activation (proliferation and migration) in an ex vivo wound healing model [65].

The fasciclin family of matricellular proteins (periostin and TGFβ-Induced Protein (TGFBI)) is known to promote wound healing partly due to increased re-epithelialization [3,44,162]. Periostin-deficient mice exhibited delayed wound healing with defective re-epithelialization [108] and myofibroblast activation that facilitates wound contraction [158,159]. Similar to the over-expression of *Thbs1* [154], constitutive overexpression of periostin dampens the wound repair process by inhibiting neutrophil and macrophage infiltration [173]. The other member of the family, TGFBI, was identified as a BM component in the epidermis. It enhances wound healing by promoting epidermal stem cell proliferation and re-epithelialization [162].

In wound healing, epidermal stem cells are empowered with plasticity that allows them to temporarily convert to other lineages (a term known as lineage infidelity) during re-epithelialization [174,175,176,177]. For example, post-injury hair follicle stem cells are mobilized to take on the role of epidermal stem cells and regenerate the epidermis [174,175]. There is still much that is unknown concerning how matricellular proteins modulate epidermal stem cells as part of the wound healing process. Among them, the R-Spondin family is known to activate Wnt signaling, which is crucial for adult epidermal stem cell maintenance [178,179]. R-Spondin 1 has been shown to regulate skin differentiation, have a predisposition for squamous cell carcinoma [160], and its topical application in rat skin promotes wound healing and re-epithelialization [161]. Interestingly, hair follicle-derived epidermal stem cells are more responsive to R-Spondin 1 and WNT7a stimulation (a characteristic of hair follicle stem cell activation) compared with native epidermal stem cells, suggesting they retain the memory from their original niches [175]. Whether matricellular proteins directly influence epidermal stem cell plasticity during injury remains unanswered and an open subject to be explored.

#### 3.1.2. Epidermal Stem Cell Aging

As we age, our skin becomes thinner and more fragile, and its wound healing ability decreases. This is partly due to the impaired epidermal regeneration process in aged skin, which suffers from imbalanced BM ECM and growth factor signaling, an increased inflammatory microenvironment, the loss of epidermal stem cell heterogeneity, and diminished stem cell potential [18,125,180,181,182,183,184,185]. During aging, epidermal stem cells proliferate less [186,187], Wnt signaling components are downregulated [112,188,189], and fast-cycling epidermal stem cells are gradually depleted, which leads to reduced tissue fitness and resilience. However, slow-cycling epidermal stem cells are maintained into old age [18]. These slow-cycling stem cells are enriched for collagen XVII expression (Figure 3) [135], enabling their firm attachment to the BM and a higher self-renewal potential. In contrast, low collagen XVII-expressing cells are outcompeted, delaminated, and differentiated in aged skin [181].

What changes occur in the BM during aging that promote the loss of epidermal stem cell heterogeneity? In young human skin, wavy, undulating structures increase the DEJ area and strengthen the connection between the epidermis and the underlying dermis to increase skin mechanical resistance [190]. In aged human skin, however, the epidermis appears flatter than in young skin [182,191]. Human epidermal stem cells residing on the different areas of the undulating BM structures are also characterized by unique mechanical properties that may contribute to their heterogeneity; cells in the tip area are softer (lower Young’s modulus) and express higher β1 integrin stem cell marker levels compared with cells at the base area [192]. Flattening of the DEJ in aging skin may affect epidermal stem cell properties and shift the population balance towards stiffer stem cell populations, similar to stem cells grown on a flat surface [192].

Although epidermal BM becomes stiffer and thicker during aging [125,193,194], the expression of multiple constituents of the BM structural ECM is decreased, including collagen IV, VII, and XVII and laminin-332 [181,194,195,196,197]. Aberrant matricellular protein expression results in matrix re-arrangement, collagen cross-linking, and dysregulated enzymatic activity, thus promoting tissue stiffness in tumorigenesis [4]. However, it is unclear whether aging also induces alterations to matricellular proteins at the BM to promote stiffness and epidermal stem cell deregulation. Here, we discuss the studies that address some aspects of this question (Table 3).

The abundance of the matricellular protein fibulin 7 (encoded by *FBLN7*) at the epidermal BM is decreased in aged skin, with a concomitant loss of epidermal stem cell heterogeneity [18]. Fibulin 7 belongs to the short fibulin family of proteins but has no elastogenic function like other family members, such as fibulin 4 and 5 [198]. Instead, fibulin 7 is essential in maintaining long-term, fast-cycling epidermal stem cell potential during chronological aging. Fibulin 7 loss of function in mice leads to early depletion of fast-cycling stem cells, as shown by lineage tracing experiments after a 1-year chase [18]. Although the molecular mechanism of fibulin 7 regulation of fast-cycling stem cells is not well-defined, fibulin 7 deletion results in higher fast-cycling stem cell proliferation at a young age, which may have contributed to earlier fast-cycling stem cell exhaustion in the aging *Fbln7* null mice compared with wild type mice. Moreover, fibulin 7 loss of function is correlated with augmented stem cell differentiation and lower collagen XVII expression, a stem cell fitness marker [18,181].

Fibulin 7 can also form a direct interaction with collagen IV, which may support its role in the BM to influence stem cell fate [18,199,200]. Other fibulin 7-binding proteins include fibronectin [18,201], which regulates cell adhesion at the BM to keep basal keratinocytes in an undifferentiated state [202,203,204], as well as periostin and tenascin C, which are both matricellular proteins highly expressed during skin inflammation and wound healing [29,79]. Further characterization of fibulin 7 function in epidermal stem cells suggests that it suppresses aging-associated inflammatory responses and lineage misregulation. Finally, fibulin 7 has demonstrated an ability to ameliorate the re-epithelialization process during full-thickness wound healing in 1-year-old, middle-aged mice, likely due to the abovementioned roles and interactions [18].

Aging-induced epidermal stem cell differentiation and hemidesmosome instability are also linked to age-related dermal stiffening and vasculature atrophy, as the induction of dermal vasculature reverses this phenotype [124]. In this study, the authors focused on an increased calcium influx in the basal keratinocytes mediated by the mechanosensitive Piezo1 ion chanel and the secretion of the immune-responsive protein pentraxin 3 by aged dermal fibroblasts, which plays an anti-angiogenic role in this context. Likewise, aged epidermal stem cells and dermal fibroblasts upregulate the Thbs1 matricellular protein, inhibiting angiogenesis [124]. This is supported by earlier studies that also found THBS1 upregulation in secreted proteins from human dermal fibroblasts isolated from intrinsically aged skin compared with young donors [164] and found that cutaneous blood flow is recovered and induced in aged *Thbs1* or *CD47* (a receptor for THBS1) null mouse skin [163]. It is still unknown whether the THBS1-CD47 axis directly regulates epidermal stem cells. However, in lung endothelial cells, THBS1-CD47 signaling inhibits self-renewal ability in cultures [205], and BMP4-dependent THBS1 expression in lung endothelial cells triggers differentiation of bronchioalveolar stem cells, which are activated during lung injury repair [206,207]. Increased THBS1-CD47 signaling has also been observed in aging human lung vasculature, promoting endothelial cell senescence [207,208].

Aging skin is associated with an increased risk of tumorigenic growth. Snail is a transcription factor known to be a driver of the EMT, a process involved in the maintenance of cancer stem cells and metastasis [209]. When Snail is overexpressed in the mouse epidermis, epidermal stem/progenitor cell populations expand, and stemness is promoted [165,166,210]. In turn, Snail induces the transcription of Mindin (SPON2) matricellular protein, which binds to integrin αMβ2 (CD11b) in an autocrine manner and activates c-Src and STAT3. Loss of function in Mindin was further shown to rescue the effect of Snail overexpression on epidermal expansion and in squamous cell carcinoma xenograft models [165]. In addition, a higher *SNAI1* expression was found in the epidermis of patients with systemic sclerosis, linking its function in skin fibrosis to the induction of inflammation and fibroblast-mediated collagen secretion [167,168]. Mindin and fibulin 5 inhibition can rescue Snail-induced skin fibrosis, making them potential therapeutic targets. As fibrosis and ECM stiffness are linked to tissue aging, it would be interesting to address the role of these proteins in the context of aging skin.

### 3.2. Hair Follicle Stem Cells

Hair follicle stem cells reside in the bulge region, which is enriched for matricellular proteins such as tenascin C, periostin, and Mindin/Spon2 [101] (Table 3). Tenascin C can interact with the Wnt3a ligand, thus facilitating Wnt signaling to maintain the hair follicle stem cell population and suppress aberrant differentiation [32]. Periostin positively regulates the proliferation of cells in the hair follicles after skin wounding and in cultured keratinocytes via activation of NFκB. The loss of periostin expression is further associated with decreased fibronectin expression around the hair follicle BM [108].

Matricellular proteins are also expressed in the dermal papilla to regulate hair follicle growth. For example, CCN2 maintains hair follicle stem cell quiescence, as its genetic deletion from the dermal papilla niche results in an increased number of hair follicles with a shorter telogen phase [109]. In contrast, the Wnt agonist R-Spondin 2, through intradermal administration, activates hair follicle stem cells and sustains the anagen phase, promoting hair growth [110]. Dermal papilla cells also express R-spondin 2 and R-spondin 3 to promote cell proliferation of hair follicle dermal stem cells that replenish the dermal sheath and hair follicle progenitors [111]. However, the inhibition of TGFβ signaling in skin fibroblasts induces dermal papilla niche reorganization, resulting in inefficient hair production and a redistribution of progenitor cells, although their proliferation and differentiation are generally preserved [211].

In aged hair follicle stem cells, *Mindin*/*Spon2* gene expression is decreased while *Thbs1* is increased [112]. The role of Mindin in hair follicle stem cells has not been reported, although it may be related to stemness, as in the case of epidermal stem cells [165]. Hair follicle stem cell activation is associated with the skin vasculature [212,213]. THBS1 is an angiogenesis inhibitor and has been shown to induce hair follicle shrinkage and vascular regression during the catagen phase of the hair cycle [113]. Thus, an increase in THBS1 expression may be linked to reduced hair density associated with aging. In line with the involution of vasculature in aged skin and dermal ECM changes [124], YAP/TAZ mechanosignaling is suppressed in aged dermal fibroblasts [127]. Conditional knockout of YAP in these fibroblasts mimics the effect of aging skin (i.e., a reduced number of hair follicles). In contrast, the expression of constitutively active YAP rescues skin from the aging phenotype (a restored number of hair follicles) [127]. Fibroblast deregulation could be further attributed to an increase in matricellular protein CCN1 in the aging dermis [169]. CCN1 augments fibroblast senescence, suppressing fibrotic genes such as *Col1a1* [149], one of the hallmarks of aging skin [214].

### 3.3. Sebaceous Gland Stem Cells

Sebaceous gland stem cells are originated from hair follicle stem cells expressing Sox9 or Lrig1 [215,216]. Sebaceous gland stem cells and progenitor cells express the embigin cell surface receptor, which binds to fibronectin at the N-terminus domain for cell adhesion. Embigin expression is regulated by Wnt signaling, and its deletion drives basal cell detachment and differentiation, resulting in increased sebaceous glands areas and an increased number of sebocytes. Intriguingly, tissue stiffness is increased in the ECM of the basal progenitor cells upon loss of embigin [58]. Tissue stiffness has been reported in skin aging, such as increased BM stiffness in the hair follicle stem cells [125], and sebaceous glands tend to become larger during aging despite reduced sebum production [217]. It has been suggested that the increased number of differentiated sebocytes due to embigin deletion is potentiated with increased age [58], indicating that an aging-dependent factor may be further modulating the embigin–fibronectin interaction.

Several signaling pathways involving Wnt, TGFβ, Shh, and Notch have been reported for homeostasis regulation and aging of the sebaceous glands [218]. Matricellular proteins such as the CCN family can bind to TGFβ, Notch, and the Wnt co-receptor LRP6, thus affecting signaling outcomes [148,219,220,221], although their roles in sebaceous glands aging have not been reported. In line with this, CCN protein expressions are changed during aging, with CCN1 upregulated and CCN2/CTGF (connective tissue growth factor) downregulated in the dermis of aged human skin [169,170,171].

### 3.4. Cross-Communications and Lineage-Fate Plasticity Potentially Involving Matricellular Proteins in the Skin Network

Cross-communications between stem cells and the surrounding cells in the niche, such as nerve cells, muscle cells, fibroblasts, immune cells, lymphatics, or vasculature, sustain skin homeostasis. The importance of matricellular proteins in maintaining such cellular cross-talks is largely unknown. For example, interactions between the hair follicle stem cells with the sympathetic nervous system secreting the neurotransmitter noradrenaline control stem cell activity. This interaction requires the arrector pili muscle, which harbors the sympathetic innervations [222]. The attachment of the arrector pili muscle to the bulge region has been attributed to the nephronectin ECM; however, matricellular proteins such as periostin, fibulin 1, and tenascin C are also present in this region, possibly stabilizing the structure [54]. Likewise, the melanocyte stem cells in the hair follicle are modulated by the sympathetic innervation. Stress causes the rush of noradrenaline and hyperproliferation of melanocyte stem cells, followed by their differentiation, eventually depleting the melanocyte stem cell pool leading to greying hair [223]. The lymphatic vasculature also functions as a hair follicle stem cell niche [224,225]. The secretome of hair follicle stem cells alters with the hair cycle; in the resting phase, they secrete angiopoietin-like protein 7 to promote lymphatic fitness and stem cell quiescence, whereas during the growing phase, netrin 4 ECM and angiopoietin-like protein 4 are increased to induce lymphatic capillary remodeling and stem cell activation [225]. Similarly, the hair cycle and hair follicle stem cells are influenced by their neighboring blood vasculature [212]. Aging hair follicle stem cells upregulate *Thbs1* expression [112], which is anti-angiogenic and may change the stem cell–vasculature interactions in aging skin. Moreover, there are defective interactions between epidermal cells and dendritic T-cells in aging skin wound healing, and overall cellular composition, transcriptomics, and cell–cell communication are compromised [180,226].

Much remains unknown regarding whether and how matricellular proteins are involved in skin stem cell regulation during wound healing and inflammation. Do matricellular proteins influence skin cell plasticity during injury-induced regeneration? Inflammation-induced skin stem cells and skin cell plasticity have been well reported [73,81,174,175,176,177,227,228,229,230,231,232]. Slow- and fast-cycling epidermal stem cells leave their unique territories and become interchangeable in wound repair [134]. Hair follicle stem cells contribute as epidermal stem cells in post-wounding epidermal regeneration. On the contrary, epidermal stem cells are described as participating in large wound-induced hair follicle neogenesis [174,176,177]. Interestingly, differentiated sebocytes expressing *Gata6* undergo de-differentiation and become epidermal stem cells during injury repair [229], and a high degree of fibroblast heterogeneity has been observed in wound healing [83,233,234]. While the cause of such plasticity or heterogeneity is linked to the upregulation of stress-associated transcription factors [174,175,227], the role of the ECM or microenvironment is unclear. In aging skin, DNA damage leads to the proteolysis of collagen XVII in the BM of hair follicle stem cells, resulting in an epidermal fate and hair loss [130], although aging-induced lineage conversion and the role of the stem cell microenvironment is still an ongoing discussion [112]. Our findings of the loss of fibulin 7 in the aging epidermal BM and its association with the loss of fast-cycling epidermal stem cells further suggest that microenvironmental alterations affect stem cell fate [18]. Matricellular proteins can influence cell behaviors [3,146]; hence, future studies that address their roles in skin stem cell regulation and cell–cell interactions would be of great interest.

## 4. Concluding Remarks

While matricellular proteins are abundantly expressed during skin development or in an inflammatory context during injury repair or disease, a basal level of expression is maintained in various compartments of the skin, which is necessary to support overall skin functioning. This occurs through matricellular proteins interacting with structural ECMs, such as collagens and elastic fibers, and through regulation of growth factor bioavailability and cell–cell or cell–matrix adhesion, thereby orchestrating cellular responses and tissue functions (Table 1, Table 2 and Table 3). The importance of matricellular protein’s spatiotemporal expression has been demonstrated in studies where constitutive null and overexpression mice have yielded identical phenotypes, adding difficulties to studying the function of these proteins in vivo. Inducible tissue-specific over-expression or loss-of-function studies would facilitate efforts to further elucidate their roles in the skin, especially in stem cell regulation. Of note, even though human and mouse skin share some similarities, they do have notable differences [235] that could limit our interpretation from studies using rodent skin.

Given the known significance of the presence of matricellular proteins during tissue development or inflammation, where various stem cells are activated, it would be unsurprising if matricellular proteins modulate stem cell niche and function during organogenesis, as well as adult stem cells and their response to injury, as shown by studies using fibulin 7 and TGFBI (Table 3) [18,162]. Nevertheless, few of the matricellular proteins that have been reported to regulate the re-epithelialization process (Table 3) link their functions to epidermal stem cells. Finally, it remains to be explored whether matricellular protein dynamics in the stem cells’ niche affect epigenetic changes involved in tissue stem cell inflammatory memory and stem cell aging [125,175,228]. This holds promise for potential future applications in regenerative medicine.

## Figures and Tables

**Figure 1 ijms-24-14274-f001:**
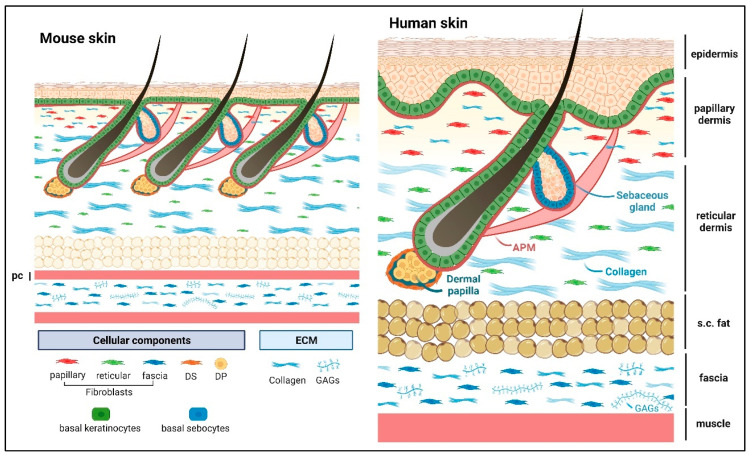
Schematic presentation of mouse and human dorsal skin. Abbreviations: DP, dermal papilla; DS, dermal sheath; pc, panniculus carnosus muscle; s.c. fat, subcutaneous fat; APM, arrector pili muscle; GAGs, glycosaminoglycans. The sweat gland is not shown in the image. Matricellular protein constituents and their roles are summarized in Table 1. Created with BioRender.com (accessed on 13 February 2023).

**Figure 3 ijms-24-14274-f003:**
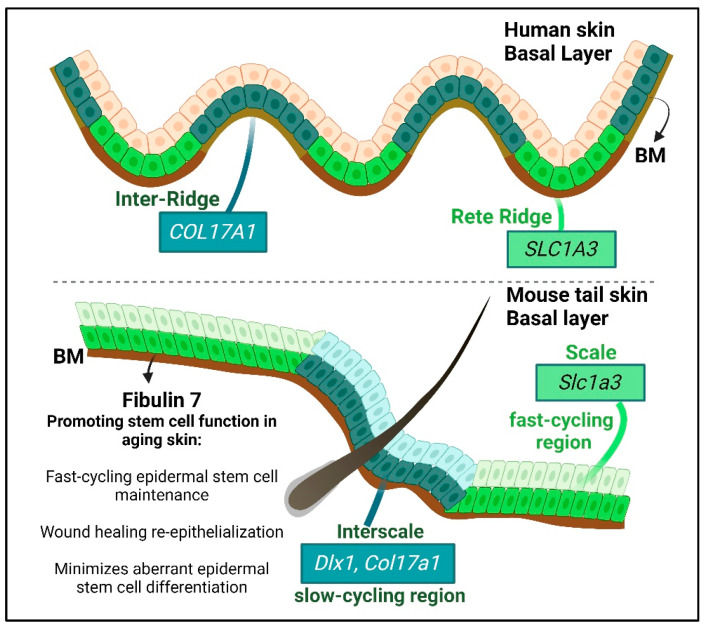
Epidermal stem cell heterogeneity in human and mouse skin. Stem cells are illustrated in dark or light green (dark green for slow-cycling stem cells and light green for fast-cycling stem cells) with some of their respective gene expression markers in boxed regions [134,135]. Fibulin 7 matricellular protein is localized in the basement membrane (BM) to support epidermal stem cell heterogeneity during skin aging. Other matricellular proteins regulating the BM, epidermal stem cells homeostasis, and the re-epithelialization process in injury-induced regeneration are summarized in Table 1 and Table 3. Created with BioRender.com (accessed on 22 July 2023).

**Table 1 ijms-24-14274-t001:** Matricellular proteins in different compartments of the skin.

Skin Compartment	Matricellular Protein	Function	References & Study Model
Epidermis	Fibulin 2	binds to laminin 332 for BM Stability	[17] (mouse)
	Fibulin 7	BM localization, binds to collagen IV in vitro	[18] (mouse)
	SPARC	binds to collagen IV in vitro and induces expression of collagen IV and VII	[19] (human skin & 3D culture), [20] (molecular structure), [21] (summary of works in mouse)
	Hemicentin 1 (Fibulin 6)	BM stability	[22] (mouse), [23] (mouse & zebrafish)
	Thrombospondin 1	inhibits angiogenesis	[24] (human skin), [25] (summary of works in mouse), [26] (mouse xenograft)
	CCN3	promotes DDR1 binding to collagen IV and inhibits melanocytes proliferation in UV-mediated stress	[27] (human skin reconstructs), [28] (summary of works in human skin & cell culture)
Dermis	Tenascin C	growth factors sequestration, regulates cell proliferation	[29] (summary), [30] (mouse NR6 cells), [31] (human skin), [32] (mouse), [33,34] (molecular interactions)
	Fibronectin *	growth factor sequestration, supports cell attachment and migration	[35] (summary), [36] (mouse)
	Fibrillin 1 *	supports elastic fiber formation and homeostasis	[37] (human & bovine tissue)
	Fibulin 2		[38] (mouse), [37] (human & bovine tissue)
			[39] (summary in human & mouse), [40,41] (human genetic mutations), [42] (mouse)
	Fibulin 4		
	Fibulin 5		
	Periostin	modulates collagen structure and stability, regulates keratinocyte proliferation	[43] (human skin), [44] (mouse skin 3D culture)
	Thrombospondin 1	dermal vascularization balance, interacts with collagen I	[25] (summary), [26] (mouse xenograft), [45] (in vivo mouse & human dermal fibroblasts)
	Thrombospondin 2	collagen structural arrangement and abundance	[46] (mouse)
	SPARC		[47] (mouse)
Dermal adipose tissue	SPARC	wound healing, modulates adipogenesis	[48,49] (mouse)
	CCDC80/URB/DRO1	modulates adipogenesis	[50] (mouse tissue and 3T3-L1 cells), [51] (human & mouse tissue), [52] (mouse)
Hair follicle	Fibulin 1	BM homeostasis	[53] (mouse), [54] [summary]
	Periostin	tendon-related genes, may provide stability for arrector pili muscle and hair follicle connection	[54,55] (mouse), [56,57] (rat)
	Tenascin C		
	SPARC		
Sebaceous glands	Fibronectin*	regulates cell differentiation	[58] (mouse)
	Tenascin C	growth factor sequestration	[59] (human & mouse eyelids)
Panniculus carnosus muscle	Fibulin 4	Panniculus carnosus muscle homeostasis	[60] (mouse)

* Fibrillin 1 and Fibronectin are not classified as matricellular proteins, but they bear non-structural functions too via binding to matricellular proteins or regulating growth factors bio-availability.

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
