# Peer review of "Matricellular Proteins in the Homeostasis, Regeneration, and Aging of Skin"

_ijms, 2023, doi:10.3390/ijms241814274_

Round 1

Reviewer 1 Report

The authors wrote a piece of information about multicellular proteins in the current manuscript entitled "Matricellular proteins in skin homeostasis, regeneration, and aging". Besides the relevant information provided, there is still a scope for improvement. Pls see my suggestions/comments below

1. Add a table for the differences between rodent and human skin

2. In the rest of the tables (#1-3), pls show/add details if anything changes for rodents comparatively in every aspect.  It is very important because most of the studies are conducted preclinically on rodents.    

Author Response

Reviewer1

The authors wrote a piece of information about multicellular proteins in the current manuscript entitled "Matricellular proteins in skin homeostasis, regeneration, and aging". Besides the relevant information provided, there is still a scope for improvement. Pls see my suggestions/comments below:

  1. Add a table for the differences between rodent and human skin

Thank you for your comments. Since differences in the skin between rodents and humans have been summarized previously, we explained it on line 583-584 and added the reference on line 588-590. Furthermore, we added additional information in Tables 1-3, explaining each reference is for rodents or humans.

  1. In the rest of the tables (#1-3), pls show/add details if anything changes for rodents comparatively in every aspect. It is very important because most of the studies are conducted preclinically on rodents.

Thank you for your suggestions. We have added the information on species, cell types and conditions wherever applicable in Tables 1-3.

Reviewer 2 Report

Having read the manuscript I have the following comments.

1. P2 Figure 1 should be placed in the text after Line 65.  Figure 2 should also appear on P3 as it mentioned on this line, it appears on P8.  Either remove reference to Fig 2 or move it from where it currently is in the manuscript.

2. P2 on L52 you mention Tables 1-3 which are on P6, 10 and 13 and not on anywhere near where they are mentioned.  Please move the table or alter this statement on L52.

3. Table 1, in he List of references cited in this table, please list references as numbers and not as author date, your reference list at the end of the manuscript is numbered and as such the references shown in the table should also be numbered.  Please correct Table 2 (P10) and Table 3 (P13) as well. 

Author Response

Reviewer 2

  1. P2 Figure 1 should be placed in the text after Line 65.  Figure 2 should also appear on P3 as it mentioned on this line, it appears on P8.  Either remove reference to Fig 2 or move it from where it currently is in the manuscript.

We have placed Figure 1 to the new position (after it is mentioned in text) and reference to Figure 2 was removed. Additionally, we have changed old Figure 3 to new Figure 2 and placed it on line 176. We also changed old Figure 2 to new Figure 3 and placed it on line 291.

  1. P2 on L52 you mention Tables 1-3 which are on P6, 10 and 13 and not on anywhere near where they are mentioned.  Please move the table or alter this statement on L52.

We removed Tables 1-3 in the statement on old line 52.

  1. Table 1, in the List of references cited in this table, please list references as numbers and not as author date, your reference list at the end of the manuscript is numbered and as such the references shown in the table should also be numbered.  Please correct Table 2 (P10) and Table 3 (P13) as well. 

We have corrected Tables 1-3. Thank you.

Round 2

Reviewer 1 Report

Accepted